# SMALL VISUAL LANGUAGE MODELS CAN ALSO BE OPEN-ENDED FEW-SHOT LEARNERS

## ABSTRACT

We present Self-Context Adaptation (SeCAt), a self-supervised approach that unlocks open-ended few-shot abilities of small visual language models. Our proposed adaptation algorithm explicitly learns from symbolic, yet self-supervised training tasks. Specifically, our approach imitates image captions in a self-supervised way based on clustering a large pool of images followed by assigning semantically-unrelated names to clusters. By doing so, we construct the 'self-context', a training signal consisting of interleaved sequences of image and pseudo-caption pairs and a query image for which the model is trained to produce the right pseudo-caption. We demonstrate the performance and flexibility of SeCAt on several multimodal few-shot datasets, spanning various granularities. By using models with approximately 1B parameters we outperform the few-shot abilities of much larger models, such as Frozen and FROMAGe. SeCAt opens new possibilities for research in open-ended few-shot learning that otherwise requires access to large or proprietary models.

## 1 INTRODUCTION

Empowered by large-scale pre-training on massive web-scraped datasets, large language models have witnessed major advancements in the past years. These large models show fascinating emergent abilities, particularly in-context learning for few-shot tasks (Brown et al., 2020; Wei et al., 2022a), which are solved without gradient-based updates, based on context samples provided via a prompt. Recently, such models have evolved from the natural language processing domain to visual language models such as Frozen (Tsimpoukelli et al., 2021) and Flamingo (Alayrac et al., 2022). Such models rely heavily on incorporating very large, proprietary language models, ranging from 7 up to 70 billion parameters, making them impractical for many individuals and organizations without access to large-scale computational resources. This paper seeks to answer whether the model scale is a crucial factor for solving open-ended few-shot learning problems.

As of yet, in-context learning has not been observed in small-scale visual language models as a mechanism for solving few-shot tasks. One reason is that these small-scale models rely heavily on semantic priors created during the pre-training and they cannot properly digest interleaved sequences of images and captions. As shown by Wei et al. (2023), if one prompts a small model with a few pairs of input-label mappings as context followed by a query sample, using new semantically-unrelated labels, the small model will stick to its semantic priors and will not adjust its predictions. Larger models, by contrast, override these priors, allowing them to learn directly from input-label mappings presented in the context, with no further gradient-based updates. This behavior is attributed to their enhanced capacity and pre-training on interleaved input-label data, which enables them to easily capture patterns and dependencies within the presented context. We hypothesize that the mechanisms for in-context learning should emerge in small models as well if we use such interleaved data and mimic the final in-context learning objective. To do so, we propose a technique to in-context learn few-shot tasks with *small* visual language models and without supervision.

We start with a small pre-trained image captioning model, using GPT-Neo (Gao et al., 2020) as a language backbone, with the aim of converting it into an in-context learner able to digest multimodal context and correctly generate a prediction for a query image. The first stage is clustering of an unlabelled image dataset, by using its embeddings and choosing a subset of the clusters. This is followed by assigning semantically-unrelated names as cluster names to the selected clusters. The

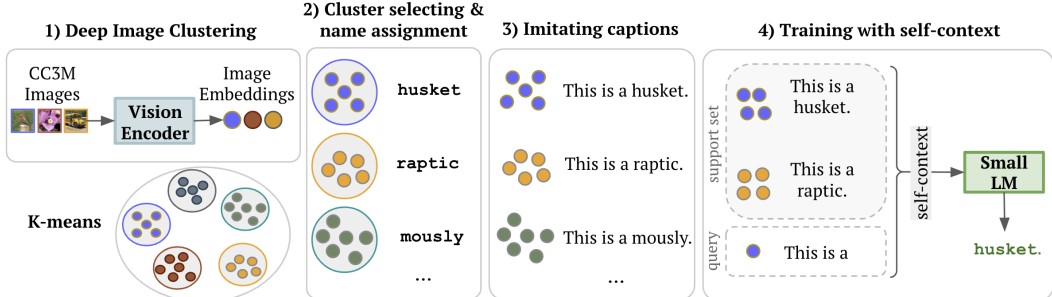

Figure 1: The SeCAt method consists of the following steps: First, the image embeddings are extracted with a vision encoder, followed by deep image clustering of the embeddings. Next is the selection of clusters and assigning arbitrary names to each one of them. Then, the assigned names are used to imitate image captions for each image in the selected clusters. The last step is the self-context adaptation of the small language model, by using the previously generated image-caption pairs.

usage of such names for clusters gives flexibility to our method because *any* word can be used for learning the patterns in a prompt. This can also be viewed as using arbitrary symbols to create a context in a self-supervised manner. Then, we imitate captions for images by converting these words into "This is a + *cluster name*" or "Here is a photo of + *cluster name*" captions, with either random or totally nonsensical meanings w.r.t. the image content. After obtaining such pseudo-captions, we construct the so-called *self-context*, which contains interleaved image-caption pairs as context and a query image. Then, we perform self-supervised fine-tuning of a pre-trained language model with mini-batches of these self-contexts, where the model is optimized to generate the correct (but semantically-unrelated) label for the query image given the context sequence. This defines our lightweight adaptation procedure, which we name *Self-Context Adaptation (SeCAt)*, and is illustrated in Figure 1.

At inference time, we keep the vision and language backbones entirely frozen and we prompt the model with multimodal contexts to perform few-shot learning. For this, we employ the multimodal few-shot datasets proposed by Tsimpoukelli et al. (2021) for fast concept binding in open-ended fashion. Furthermore, to test the ability of the model to deal with different levels of task granularity, we also evaluate our approach on semantically-easy and hard few-shot tasks based on five common vision datasets. With this, we show that the flexibility of constructing self-contexts provides the opportunity to control the difficulty and granularity of the few-shot tasks. Last but not least, we show that SeCAt can turn even small visual language models, of the order of 1B parameters, into strong in-context learners for open-ended few-shot learning, without any *supervised* fine-tuning.

To summarize, we contribute in three major aspects: *Conceptual:* We present an efficient framework for unlocking in-context learning in small visual language models for open-ended few-shot learning. *Methodological:* We introduce a self-supervised adaptation algorithm to learn an in-context template with the semantically-unrelated words for small visual language models. *Empirical:* We conduct extensive experiments on several multimodal few-shot datasets ranging from coarse to fine-grained tasks, and show that we achieve better performance compared to the larger counterparts.

## 2 RELATED WORK

**Few-shot Learning in Language Models.** Large language models have garnered substantial attention within the NLP community (Brown et al., 2020; Chan et al., 2022; Chowdhery et al., 2022; Dai et al., 2019; Tan et al., 2020; Yang et al., 2022) due to their capacity to generate extensive text as well as their remarkable in-context capabilities. Achieving this, often requires scaling transformer-based models (Rae et al., 2021; Smith et al., 2022; Chowdhery et al., 2022), augmenting pre-training data (Hoffmann et al., 2022), and advanced loss functions (Wei et al., 2021; Tay et al., 2022). The in-context learning paradigm was first introduced by GPT3 (Brown et al., 2020) as a *training-free* learning framework for few-shot learning. Numerous works have further explored this ability and showcased that it makes it easier to incorporate outside knowledge into language models by changing the context and templates (Liu et al., 2021; Wei et al., 2022b; Lu et al., 2021), and exploit it as

an interpretable interface to communicate with large language models (Brown et al., 2020). Yet, the emergent in-context learning ability comes with the cost of a huge number of parameters and a large-scale pre-training dataset. For instance, GPT3 consists of 175B parameters and is trained on approximately 45TB of text data. Alternatively, methods like MetaICL (Min et al., 2022) aim to fine-tune a smaller language model for in-context learning and evaluate it on a large set of language classification tasks, without involving text generation. Different from these works, we propose an algorithm that unlocks in-context learning in small visual language models for open-ended few-shot learning.

**Multimodal Few-shot Learning.**  Recent advancements in vision and language have arisen with the emergence of large language models (Radford et al., 2021; Ramesh et al., 2021; Saharia et al., 2022; Alayrac et al., 2022; Jia et al., 2021; Hao et al., 2022; Najdenkoska et al., 2023; Wang et al., 2022). We highlight Flamingo (Alayrac et al., 2022), FROMAGe (Koh et al., 2023), and ClipCap (Mokady et al., 2021) as notable examples. In these works, the in-context ability emerges by scaling up the number of transformer parameters, which has previously proven effective in various NLP tasks. Additionally, several methods, including Flamingo, FROMAGe, MetaLM (Hao et al., 2022), and KOSMOS-1 (Huang et al., 2023), incorporate interleaved sequences of images and captions during training. This approach simulates few-shot learning scenarios, enabling large language models to capture patterns among multiple image-caption pairs within a single sequence, thereby facilitating few-shot learning. It is important to note that ClipCap (Mokady et al., 2021) does not exhibit the in-context learning mechanism as it is not trained on interleaved images and captions. Similar to FROMAGe and Flamingo, our method benefits from interleaved sequences of images and text during training, while we differ in language model size, pre-training dataset size, and the use of distinct loss functions during the adaptation phase. Despite our focus on small-scale visual language models, we still enable in-context learning capabilities for multimodal few-shot learning problems.

**Unsupervised Pseudo-label Generation.**  Generating pseudo-labels by clustering has proven effective in unsupervised representation learning (Asano et al., 2020b; Caron et al., 2018; Ji et al., 2019; Van Gansbeke et al., 2020; Xie et al., 2016; Yang et al., 2016). This approach involves using pseudo labels in the visual domain for tasks such as image representation learning (Caron et al., 2018; Bojanowski & Joulin, 2017; Noroozi et al., 2018), image segmentation (Melas-Kyriazi et al., 2022), and video understanding (Asano et al., 2020a; Gavrilyuk et al., 2021). A self-labeling method is proposed in Asano et al. (2020b), driven by k-means and repurposed to learn a shared set of labels between audio and text modalities. Inspired by this work, we propose a self-supervised approach using k-means clustering to assign semantically-unrelated words as labels to the visual clusters and then imitate interleaved sequences of image-caption pairs based on these labels.

## 3 METHODOLOGY

To enable open-ended few-shot learning via in-context mechanisms, we propose a self-supervised adaptation technique that mimics the final in-context learning objective, but does not rely on any labeled or captioned data.

At a high level, our method clusters a large pool of images to identify highly coherent groups and assigns them names that are meant to not necessarily fit or describe the content. This noisy set of images and names is then used for adapting the model in a manner that simulates in-context learning. Our method allows for controlling the context difficulty by sampling items from distant or close clusters and by doing so it allows the final model to work well even for fine-grained few-shot learning. In the next sections, we will first formally state the problem, then describe the procedure for generating the self-supervised image-caption pairs, then we will outline the construction of self-context training samples and how we vary their difficulty, as well as the final training procedure.

**Problem statement.**  Few-shot in-context learning aims to generate the correct caption $t_q$ corresponding to a query image $x_q$ given samples of paired images $x_s$ and captions $t_s$ in a support-set $s \in \mathcal{S}$, handled by a visual language model $f$, defined as follows:

$$f(\{(x_s, t_s)\}_{s \in \mathcal{S}}, x_q) = t_q. \tag{1}$$

In order for the model to "learn" from the context, the support-set $\mathcal{S}$ contains a similar image as the query. More specifically, in the case of utilizing a language model as a decoder, the task is

"open-ended", *i.e.*, $t_q$ must be obtained via text generation, and not via classification into a fixed set of labels. Naturally, we can train a visual language model with this objective, be it that this requires access to a set of paired image-text data, as we can see from eq. (1). Instead of obtaining supervised sets of image-caption pairs, we propose to mimic this data using self-supervision and use the generated image-text pairs to finetune the visual language model.

**Model overview.**  The architecture that we use is based on image captioning encoder-decoder models. Note that in these models, such as ClipCap (Mokady et al., 2021), $f$ is the model that first embeds an image with a vision encoder $\Psi$ and then maps it into the representation space of a language model (LM), *i.e.*, $f=\mathrm{LM}(\Psi(x))$. To perform this mapping, it uses a mapping function implemented as a simple multi-layer perceptron, which outputs the visual embeddings as a visual prefix for the language model.

### 3.1 Generating Image-Pseudo Caption Pairs

**Imitating image labels.**  Let $h : x \to c$ define the human annotation process of classifying an image $x$ in a dataset $\mathcal{X}$ into class $c \in \mathcal{C}$ of a classification system $\mathcal{C}$. We replace $h$ by a composition of two unsupervised functions, $h \approx c \odot m$. The first component $c$, first clusters the dataset $\mathcal{X}$ in a self-supervised manner. For this, we utilize the visual embeddings obtained by a visual encoder $\Psi$ and cluster the whole dataset, defined as:

$$c(x) = K\text{-means}[\{\Psi(x')\}_{x' \in \mathcal{X}}](x), \tag{2}$$

where $K$ is the number of clusters and the resulting output of $c$ indicates the cluster ID for a given image. Next, we assign each ID to an arbitrary label to obtain the paired data.

**Imitating image captions.**  To arrive at pairings of captions to a given image cluster $c$, we utilize a vocabulary of words $w \in \mathcal{V}$, that are supposed to not contain words semantically related to the image (indeed, we show that a list of random names suffices for this). Next, we utilize the visual language model $f$ for the cluster name assignment, i.e. the matching step. To match the words with clusters, we embed with $\Psi$ one exemplar image per cluster, namely the cluster centroid, and embed the vocabulary words into their language model token-space using the tokenizer-embedding function $\tau$. Note that now, both $\Psi(x)$ and $\tau(w)$ are in the same embedding space, so we can simply construct a similarity matrix $\mathbf{S} \in \mathbb{R}^{K \times |\mathcal{V}|}$ by computing their cosine-similarities:

$$\mathbf{S} = \mathrm{sim}(\Psi(x), \tau(w)). \tag{3}$$

Finally, we match each image cluster with a word embedding by using the Kuhn-Munkres (Hungarian) algorithm (Kuhn, 1955) to minimize the overall cost. The matching algorithm takes this output and yields the assigned word given a cluster ID. Afterward, the captions are imitated by converting these cluster names into "This is a + *cluster name*" captions (note that other templates are also possible) and are paired with all images belonging to the particular cluster.

### 3.2 Self-Context Construction

To construct an interleaved sequence of self-context samples, we randomly pick images according to their cluster membership, during the mini-batch construction. By choosing the level of similarity between two or more clusters, from which the support set is constructed, we can control the difficulty of the problem. This provides the flexibility of the model to be adapted for more specific datasets, usually with more fine-grained data samples. For a given cluster $k$, we then sample items $(x_i, t_i)$ s.t. $c(x_i)=k$, which represent an image-caption pair belonging to the self-context. Figure 2 illustrates how the self-context is constructed.

Optionally, we can vary the difficulty of the few-shot tasks depending on the proximity between cluster centroids. This means that if two clusters are far away from each other, they create an *easy* self-context. Contrary, if they are close they create a *hard* self-context since the image samples from closer clusters have potentially more visual similarities between each other, rather than distant clusters.

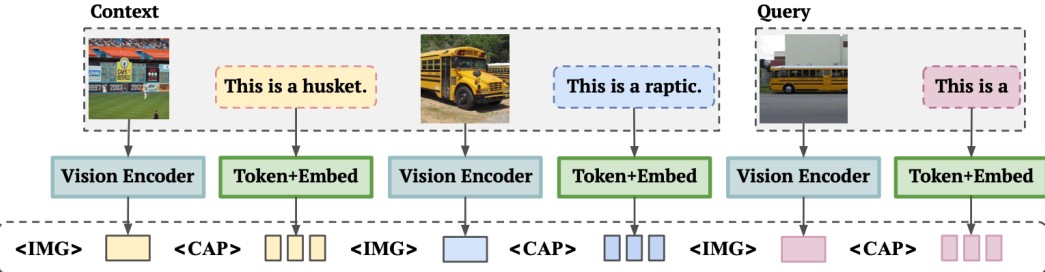

Figure 2: The self-context is represented as a sequence of interleaved pairs of images and pseudo-captions. It uses special tokens such as `` and `<CAP>` to denote the position of the elements in the sequence and is used as an input to the language model to complete the sentence for the query image. Note that this is an example of a 2-way 1-shot self-context sequence.

## 3.3 MIXED SELF-CONTEXT LEARNING & INFERENCE

Given the image-caption mappings $(x_s, t_s)$ as a self-context, and the query image $x_q$, the learning process is performed by optimizing the cross-entropy loss, while generating the query caption $t_q$, as:

$$\mathcal{L} = H(f(\{(x_s, t_s)\}_{s \in \mathcal{S}'}, x_q)|t_q). \tag{4}$$

Note that the loss function uses the constructed self-context as a single data point. To encourage generalization to different context lengths with one model we perform *mixed* self-context learning, where we randomly vary the context length within a batch. This means that we change the number of samples in the context by taking into account 2-way and $j$-shot tasks alternately, where $j \in \{1, 3, 5\}$.

At inference time, we keep the full model entirely frozen, and we test its ability to digest new in-context sequences. We consider previously unseen few-shot tasks, which also have a support set as a context, and a query sample to evaluate the performance. Specifically, the model completes the sentence for each query sample in an open-ended autoregressive manner. To obtain the final output, we use beam-search to sample from the language model given the sequence of context samples.

## 4 EXPERIMENTS

### 4.1 EXPERIMENTAL SETUP

**Datasets.** To pre-train an image captioning model and to perform the clustering part, we use the *Conceptual Captions (CC3M)* dataset (Sharma et al., 2018), which consists of 3M pairs of images and captions, web-scrapped and post-processed. At the inference stage, we employ multimodal few-shot datasets (Tsimpoukelli et al., 2021), namely *Real-Names miniImageNet* and *Open-Ended miniImageNet*, each one with 1, 3 and 5 shots, with 2 and 5-way tasks. The evaluation setting is similar to MetaICL (Min et al., 2022) which also investigates in-context abilities but only for text classification.

Additionally, to test the ability of our approach to generalize across fine-grained and coarse-grained settings, we create semantically *easy* and *hard* datasets. In particular, we reorganize existing datasets, namely *OxfordPets* (Parkhi et al., 2012), *Flowers102* (Nilsback & Zisserman, 2008), *Food101* (Bossard et al., 2014), *CUBS-200* (Wah et al., 2011) and *SUN397* (Xiao et al., 2010). For the semantically-easy split, given an $n$-ways $k$-shots scenario, we randomly choose $n$ datasets from the pool of these five datasets. Subsequently, from each selected dataset, we randomly select a single class to constitute the $n$-ways setting. For the semantically-hard split, we randomly select one dataset, followed by the selection of $n$ classes from that chosen dataset. Finally, from the chosen classes, we randomly select $k$-image samples. We provide more details about the construction of the easy and hard-splits in the appendix. The splits will be released to foster further study.

**Implementation details.** The language backbone of our model is based on the GPT-family of models, namely GPT-Neo model (Gao et al., 2020), and the smaller versions, GPT2-small and GPT2-medium. We utilize the vision encoder from the pre-trained CLIP ViT-B/32 model (Radford et al., 2021) for our model's visual backbone. To ensure that the model correctly pays attention to the image

Table 1: Baselines comparison on 2- and 5-way Real-Name miniImageNet and Open-Ended miniImageNet in accuracy(%). Note that OpenFlamingo (Awadalla et al., 2023) is considered an upper-bound. Our SeCAt-trained model outperforms its counterparts using up to 5× fewer parameters.

| | | Real-Name miniImageNet | | | | Open-Ended miniImageNet | | | |
| | | 2-way | | 5-way | | 2-way | | 5-way | |
| Methods | #params | 1-shot | 5-shot | 1-shot | 5-shot | 1-shot | 5-shot | 1-shot | 5-shot |
|---|---|---|---|---|---|---|---|---|---|
| ClipCap (Mokady et al., 2021) | 1.3B | 0.0 | 0.0 | 0.02 | 0.0 | 0.0 | 0.0 | 0.0 | 0.01 |
| Frozen (Tsimpoukelli et al., 2021) | 7B | 33.7 | 66.0 | 14.5 | 33.8 | 53.4 | 58.9 | 51.1 | **58.5** |
| FROMAGe (Koh et al., 2023) | 6.7B | 31.0 | 50.4 | 17.5 | 30.7 | 27.8 | 49.8 | 16.3 | 19.5 |
| **SeCAt (Ours)** | 1.3B | **85.7** | **83.2** | **68.6** | **58.0** | **87.4** | **85.6** | **68.0** | 41.9 |
| OpenFlamingo (Awadalla et al., 2023) | 9B | 62.0 | 95.9 | 45.3 | 91.2 | 45.2 | 63.4 | 15.0 | 56.9 |

Table 2: Generalization from easy-to-hard on the 2- and 5-way Easy vs Hard dataset splits in accuracy(%). Note that OpenFlamingo (Awadalla et al., 2023) is considered an upper-bound. Our SeCAt-trained model better adjusts to easy-to-hard dataset splits than 5× larger FROMAGe model.

| | | Easy split | | | | Hard split | | | |
| | | 2-way | | 5-way | | 2-way | | 5-way | |
| Methods | #params | 1-shot | 5-shot | 1-shot | 5-shot | 1-shot | 5-shot | 1-shot | 5-shot |
|---|---|---|---|---|---|---|---|---|---|
| ClipCap (Mokady et al., 2021) | 1.3B | 0.0 | 0.0 | 0.0 | 0.0 | 0.0 | 0.0 | 0.0 | 0.0 |
| FROMAGe (Koh et al., 2023) | 6.7B | 30.0 | 50.1 | 13.8 | 28.3 | 28.6 | 46.6 | 10.0 | 23.5 |
| **SeCAt (Ours)** | 1.3B | **81.3** | **65.2** | **70.5** | **49.7** | **63.8** | **52.6** | **34.7** | **26.2** |
| OpenFlamingo (Awadalla et al., 2023) | 9B | 53.3 | 98.9 | 37.8 | 98.8 | 39.9 | 90.3 | 25.9 | 78.0 |

and caption during training, we add special tokens `` and `<CAP>` in the prompt before the image and caption respectively (see Figure 2). We have found this to be particularly useful for in-context learning because it helps the language model to focus on attending to the correct image and text within the interleaved prompt sequence. To implement the deep clustering stage, we use the Faiss library (Johnson et al., 2019), particularly the k-means algorithm with 10 iterations. The full implementation is in PyTorch and HuggingFace (Wolf et al., 2020) and will be publicly released. We provide additional details on the implementation and hyperparameters in the appendix.

**Training details.** Our models are trained using mixed-precision with Bfloat16 (Abadi et al., 2016). In the image captioning pre-training stage, we use a batch size of 160 over 370,000 iterations and 3 A6000 GPUs. Furthermore, we use the AdamW optimizer (Kingma & Ba, 2016) with a learning rate of $2e$-5 and a warmup of 5000 steps. We set the visual prefix length to 5 and the word embedding dimension to 2048. During the self-context adaptation stage, we only fine-tune the language backbone with a small learning rate of $5e$-6 for 50 epochs and keep all other components fixed.

**Evaluation criteria.** We evaluate our approach in an open-ended fashion, by measuring the accuracy(%) of generating the words which match the ground-truth.

## 4.2 RESULTS & DISCUSSION

**Baseline comparison.** In multimodal few-shot learning scenarios, fast concept binding pertains to the ability of the model to learn the connection between visual concepts and words by observing only a few demonstrations. The experiments in Table 1, measure to what extent our SeCAt approach is able to perform such binding with language models of 1.3B parameters. Our experiments cover 2 and 5 ways, each one with 1 and 5 shots. It can be observed that our approach outperforms models that are up to 5× larger, such as Frozen (Tsimpoukelli et al., 2021) and FROMAGe (Koh et al., 2023). This shows that small models can indeed be adapted to be good in-context learners for few-shot learning in a fast and efficient manner. We view OpenFlamingo (Awadalla et al., 2023) as an upper-bound of our approach since it is pre-trained on web-scraped interleaved sequences of images and text, which directly helps in-context learning abilities. While OpenFlamingo employs 5× more parameters and trains on extensive datasets such as LAION2B (Schuhmann et al., 2022) (with 2B image-text pairs) and Multi-modal C4 (Zhu et al., 2023) (with 104M combined image-text samples), our method bypasses such extensive pre-training by leveraging our unique self-supervised approach.

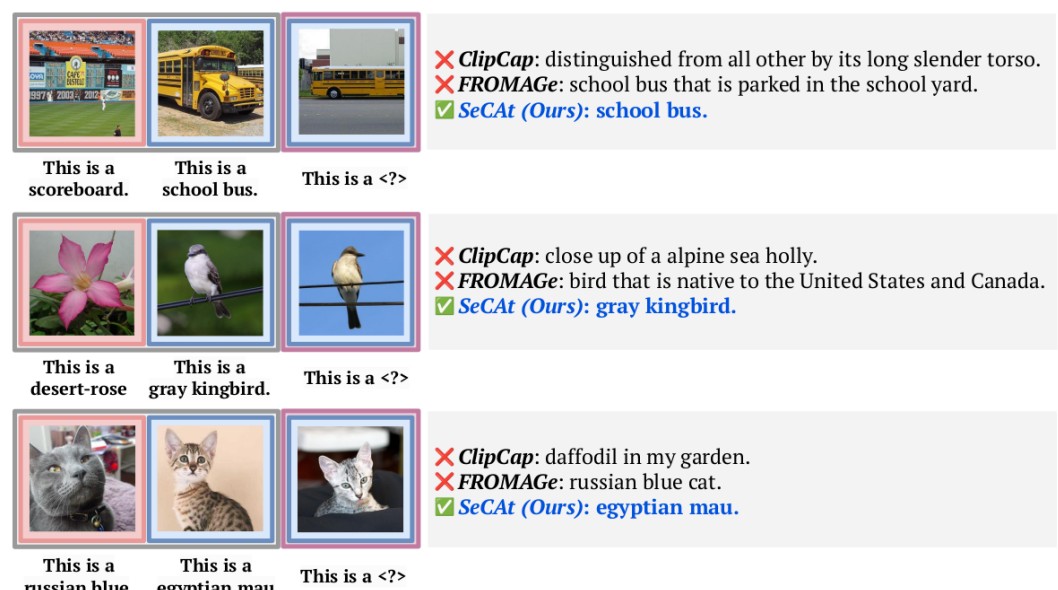

Figure 3: Qualitative comparison between SeCAt and two other baselines, ClipCap and FROMAGe, on a 2-way 1-shot tasks from Real-Names miniImageNet (first row), Easy-split (second row) and Hard-split (third-row). SeCAt generates the prediction correctly by following the concept binding in the context sequence, whereas ClipCap produces nonsensical words and FROMAGe generates excessively detailed captions of the image, showing that they are unable to learn from the context.

**Generalization from easy-to-hard.** The flexibility of our approach, to select clusters with a particular distance and label them in a self-supervised manner, allows to handle both fine-grained and coarse-grained few-shot tasks. In Table 2, we demonstrate the performance on easy and hard-splits, which are defined in Section 4.1, revealing the ability of our approach to adapt to different levels of task difficulty. As expected, it is easier for the model to adjust to the easy-split settings, compared to the hard-split. Similarly as in Table 1, our SeCAt approach outperforms FROMAGe (Koh et al., 2023), across all settings, even though it is using a notably smaller language model, meaning that visual language models can indeed benefit from having SeCAt as an efficient adaptation step.

**Qualitative analysis.** In Figure 3, we show examples of a 2-way 1-shot tasks, with an interleaved image-caption sequence and a query image. We analyze the generated output obtained by feeding this sequence in our SeCAt model and the baseline models ClipCap and FROMAGe. It can be seen that SeCAt successfully binds visual concepts in the image to the relevant words, and is able to produce the expected output. Contrary to this, ClipCap generates an incorrect caption, not related to the query image, showing the lack of in-context learning ability in small visual language models without SeCAt. Interestingly, FROMAGe is able to capture the concept of *school bus* or *bird* as predictions, but it is also excessively verbose. This essentially means that it is leveraging its semantic priors from the image captioning pre-training and not entirely adapting to the context sequence. Similar observations are present in other qualitative results, which we provide in the appendix.

### 4.3 ABLATIONS

In the next sections, we ablate our method on the Real-Name miniImageNet dataset using 2-way and 5-ways in both 1-shot and 5-shot settings.

**Effect of self-context difficulty.** Our method is sufficiently flexible to vary the difficulty of the self-context construction. We can use cluster centroids, in close proximity or further apart from each other, to influence the semantics of the chosen visual concepts within the self-context. We consider three different settings by computing L2 distances between all centroids. The *hard* setting takes the most similar 5%, the *easy* setting takes the least similar 5%, and the *varying* setting shuffles the clusters from both the hard and easy settings. As can be observed from Table 3a, the hard setting

Table 3: Ablations. We ablate the key components of our method, namely (a) Effect of self-context difficulty, (b) Influence of semantically-unrelated names, (c) Matching names to cluster centroids, (d) Benefit of mixed self-context training, (e) Impact of language model size, and (f) Generalization on different prompt templates. Evaluations are done on the 2- and 5-way Real-Name miniImageNet with the best model from Table 1.

(a) Effect of self-context difficulty.

| | 2-way | | 5-way | |
|---|---|---|---|---|
| difficulty | 1-shot | 5-shot | 1-shot | 5-shot |
| hard | 32.6 | 39.4 | 14.9 | 8.6 |
| easy | 82.2 | 81.8 | 52.5 | 29.8 |
| varying | **85.7** | **83.2** | **68.6** | **58.0** |

(b) Influence of semantically-unrelated names.

| | 2-way | | 5-way | |
|---|---|---|---|---|
| vocabulary | 1-shot | 5-shot | 1-shot | 5-shot |
| nonsense | 77.2 | 69.7 | 55.7 | 10.3 |
| numbers | 81.6 | 54.8 | 49.4 | 24.9 |
| nouns | **85.7** | **83.2** | **68.6** | **58.0** |

(c) Matching names to cluster centroids.

| | 2-way | | 5-way | |
|---|---|---|---|---|
| matching | 1-shot | 5-shot | 1-shot | 5-shot |
| random | 81.8 | 83.2 | **68.7** | 40.7 |
| cost-based | **85.7** | **83.2** | 68.6 | **58.0** |

(d) Benefit of mixed self-context training.

| | 2-way | | 5-way | |
|---|---|---|---|---|
| setting | 1-shot | 5-shot | 1-shot | 5-shot |
| single-task | 73.3 | 25.1 | 35.2 | 3.6 |
| mixed-task | **85.7** | **83.2** | **68.6** | **58.0** |

(e) Impact of language model size.

| | 2-way | | 5-way | |
|---|---|---|---|---|
| LM | 1-shot | 5-shot | 1-shot | 5-shot |
| GPT2$_{small}$ | 26.9 | 54.3 | 37.5 | 33.1 |
| GPT2$_{medium}$ | 56.2 | 64.2 | 42.4 | 41.7 |
| GPT-Neo | **85.7** | **83.2** | **68.6** | **58.0** |

(f) Generalization on different prompt templates.

| | 2-way | | 5-way | |
|---|---|---|---|---|
| Template | 1-shot | 5-shot | 1-shot | 5-shot |
| "A photo of a" | 73.0 | 70.3 | 45.0 | 43.2 |
| "On this picture there is a" | 72.5 | 67.8 | 58.2 | 39.2 |
| "This is a" | **85.7** | **83.2** | **85.7** | **58.0** |

performs considerably worse than the other two, as the model deals with images clustered closely together with limited variability. For both the easy and varying settings the performance increases. We conclude that our approach benefits from varying the proximity between cluster centroids.

**Influence of semantically-unrelated names.** For the selection of the semantically-unrelated names used for labeling the clusters and then generating the pseudo-captions of images, we consider either nonsense words, random numbers, or random nouns. The nonsense words are taken using a nonsense-word generator[1], similar to Tsimpoukelli et al. (2021). The random numbers and nouns are generated in a similar manner and are semantically-*unrelated* to the clustered images. Table 3b shows the performance per vocabulary choice, across different few-shot settings. The random nouns yield better performance than the random numbers and nonsense names. Even though the cluster names are unrelated to the images in the cluster, the model still achieves satisfactory performance. This suggests that any word embedding is good enough for the model to learn since it views them as mere symbols helpful for learning a self-context pattern.

**Matching names to cluster centroids.** The impact of the name-matching techniques is explored in Table 3c, where we compare random cluster-name matching and cost-based matching. In the random cluster-name matching variant, the name embeddings are randomly assigned to cluster centroids. The cost-based matching variant utilizes the Kuhn-Munkres (Hungarian) algorithm Kuhn (1955), which aims to find the minimal distance between cluster centroids and name embeddings. The cost-based matching approach yields better performance, which means that SeCAt benefits from a more informed manner of cluster naming.

**Benefit of mixed self-context training.** To evaluate the influence of varying self-context length, we consider two adaptation strategies. The first strategy, denoted as single-task, is simply using a fixed number of samples in the self-context across all mini-batches, where we consider only 2-way 1-shot tasks. The second strategy is the mixed self-context training, where we randomly vary the number of

---

[1] https://www.soybomb.com/tricks/words/

samples by using 2-way and $j$-shot tasks, where $j \in \{1, 3, 5\}$. Comparing the two strategies in Table 3d reveals that mixed self-context training consistently outperforms the single one by a considerable margin, especially when the number of shots increases. This is mainly attributed to the fact that the mixed training paradigm lets the model observe different lengths of the self-context sequences.

**Impact of language model size.** To investigate the impact of the language model, we replace the GPT-Neo backbone (1.3B parameters) with its smaller alternatives GPT2-medium (355M parameters) and GPT2-small (124M parameters) and report results in Table 3e. Naturally, the best performance is obtained with the largest variant, but the two smaller alternatives also show satisfying results, especially if we take into account the considerable difference in size. We looked at the training times required for each variant of the language backbone. The best version of our approach using GPT-Neo can be trained in just 14 hours, unlike larger variants which require more than a day of training (e.g. FROMAGe (Koh et al., 2023)). Moreover, training the smaller variants is even faster: 6 hours for GPT2-small and 11 hours for GPT2-medium. This time efficiency is crucial when rapid model adaptation is necessary or when access to large models and computational resources is limited.

**Generalization on different prompt templates.** We adapt the model using the common prompt "This is a + *label*" and report results based on it. To demonstrate the robustness of our model to other prompts at inference time, we introduce alternative prompt templates. Particularly, we use: "A photo of a + *label*" and "On this picture, there is a + *label*". The performance, as presented in Table 3f, affirms our method's strong generalization across varied prompt templates, negating the possibility of overfitting to a specific prompt.

**Influence of the varying number of clusters.** The number of clusters can be tuned depending on the fine-graininess of the problem at hand. We evaluate our best setting by using different numbers of $k$ clusters, where $k \in \{25, 50, 75, 100, 200\}$. As can be seen in Figure 4, we observe a consistent increase in the performance of up to 100 clusters. We assume that, as the miniImageNet evaluation datasets are not fine-grained enough, the performance slightly starts degrading for $k = 200$.

**Limitations.** Our work aims to unlock in-context learning in small visual language models for open-ended few-shot learning. It achieves the necessary capacities to some degree, but it can benefit from extending the evaluation on more complicated tasks which can give a clearer picture of possible applications. However, it is already able to achieve good performance on open-ended few-shot learning, which can be easily extended to other open-ended vision-language tasks, such as image captioning and visual question answering as future work.

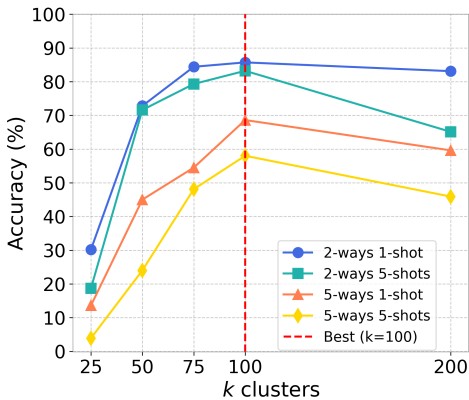

Figure 4: Influence of varying numbers of clusters for generating the pairs of images and pseudo-captions. The accuracy increases up to 100 clusters.

## 5 CONCLUSION

We introduce Self-Context Adaptation (SeCAt), a self-supervised learning approach able to unlock in-context learning abilities in small visual language models for open-ended few-shot learning. It does so, by leveraging clustering to group unlabelled images and assign semantically-unrelated names to these clusters, simulating image captions. This yields sequences of self-contexts which are used as inputs to the language model to further adapt it to easily capture patterns and dependencies within the presented context. Our experiments confirm that SeCAt can teach models how to digest multimodal contexts, even by using models that do not immediately exhibit in-context learning abilities. Last, but not least, our approach demonstrates efficiency in terms of data and training resources, contributing to the advancement of multimodal few-shot learning that is otherwise closed to individuals without access to large, proprietary models.

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
