# SMALL VISUAL LANGUAGE MODELS CAN ALSO BE OPEN-ENDED FEW-SHOT LEARNERS

## APPENDIX

The supplementary materials consist of the following sections: A. Construction of Easy and Hard splits, B. Hyperparameters details, C. Additional quantitative evaluation, D. Additional qualitative evaluation.

## A    CONSTRUCTION OF EASY AND HARD-SPLITS

To construct the Easy and Hard-splits for additional evaluation we use with five different datasets, such as fine-grained classification ones i.e. *OxfordPets* (Parkhi et al., 2012), *Flowers102* (Nilsback & Zisserman, 2008), *Food101* (Bossard et al., 2014), *CUBS-200* (Wah et al., 2011) and scene understanding i.e. *SUN397* (Xiao et al., 2010). The procedure is somewhat similar to the one in Frozen (Tsimpoukelli et al., 2021) for creating multimodal few-shot benchmarks. All images are taken from the training partitions of the mentioned datasets.

For the Easy-split, to generate a 2-way task with $n$ shots, the following steps are employed:

1. Sample two datasets d1, d2 from the pool of five datasets.

2. From each dataset, sample 1 class, namely c1 from dataset d1 and c2 from dataset d2.

3. Sample $n$ images $[x_1^{c_1}, \ldots x_{n+1}^{c_1}]$ from c1, and n images $[x_1^{c_2}, \ldots x_n^{c_2}]$ from c2. Note that from one class (e.g. c1) we sample $n+1$ images since we use one sample as a query.

4. Prepend the truncated caption "This is a " to the labels c1 and c2, to obtain the full caption $y_1$ and $y_2$ respectively for the images.

5. Interleave the pairs of images and captions in a sequence, as follows: $[(x_1^{c_1}, y_1), (x_1^{c_2}, y_2), \ldots (x_n^{c_1}, y_1), (x_1^{c_2}, y_2)]$.

For the Hard-split, to generate a 2-way task with $n$ shots, the following steps are employed:

1. Sample one dataset d1 from the pool of five datasets.

2. From the selected dataset, sample 2 different classes, namely c1 and c2.

3. Sample $n$ images $[x_1^{c_1}, \ldots x_{n+1}^{c_1}]$ from c1, and n images $[x_1^{c_2}, \ldots x_n^{c_2}]$ from c2. Note that from one class (e.g. c1) we sample $n+1$ images since we use one sample as a query.

4. Prepend the truncated caption "This is a " to the labels c1 and c2, to obtain the full caption $y_1$ and $y_2$ respectively for the images.

5. Interleave the pairs of images and captions in a sequence, as follows: $[(x_1^{c_1}, y_1), (x_1^{c_2}, y_2), \ldots (x_n^{c_1}, y_1), (x_1^{c_2}, y_2)]$.

To generate 5-way tasks, the above process is generalized. Particularly, for the Easy split, in step 1 we consider all five datasets, without sampling a subset of them. On the other hand, for the Hard split in step 2, we sample 5 different classes within a dataset to obtain the 5-ways.

## B    HYPERPARAMETERS DETAILS

We provide the detailed hyperparameters used for the image captioning pre-training stage and self-context adaptation stage in Table 1. Regarding the computational resources, for the pre-training stage,

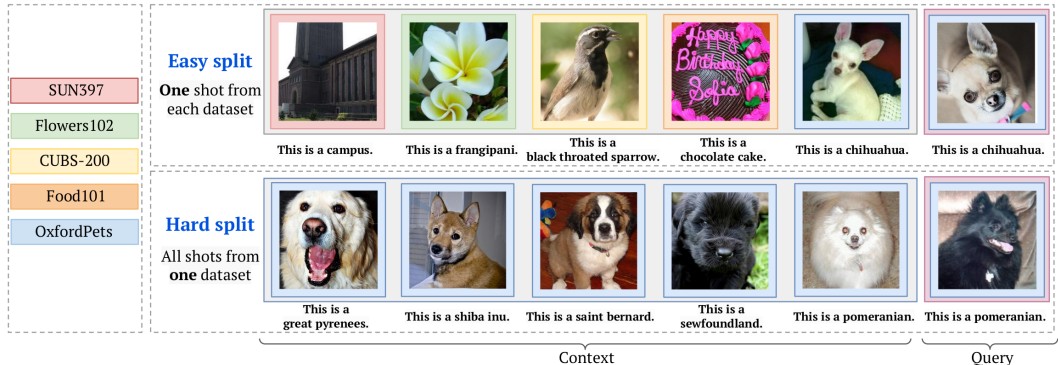

Figure 1: Examples of the restructured datasets to obtain the *easy* and *hard* few-shot tasks. The top row illustrates a 5-way 1-shot task from the easy split, with a shot per dataset. The bottom row depicts a 5-way 1-shot task from the hard split, where all shots are selected from one dataset.

we use three A6000 GPUs for the duration of four days, and for the self-context adaptation stage, we use one A6000 GPU for the duration of 14 hours in case of using the GPT-Neo. For the smaller version of the language models, the pre-training time takes around two days for GPT2-medium and one day for GPT2-small. The self-context adaptation takes 11 hours for GPT2-medium and 6 hours for GPT2-small.

Table 1: List of hyperparameters used to reproduce the experimental results in the paper, both for the pre-training and self-context adaptation stage.

| Hyperparameters | Pre-training | Self-context adaptation |
|---|---|---|
| LM choice | GPT-Neo, GPT2-small(-medium) | GPT-Neo, GPT2-small(-medium) |
| Vision encoder | CLIP ViT-B/32 | CLIP ViT-B/32 |
| Optimizer | AdamW | AdamW |
| Learning rate (LR) | $2e$-5 | $5e$-6 |
| LR scheduler | Linear | Linear |
| Batch size | 160 | 15 |
| Iterations | 370,000 | 80,000 |
| Warm-up steps | 5000 | 5000 |
| Visual prefix length | 5 | 5 |
| Word embedding size | 2048 (768, 512) | 2048 (768, 512) |
| Images size | $224 \times 224$ | $224 \times 224$ |
| Sequence padding | 10 | 10 |
| Few-shot tasks | N/A | 10,000 |

## C ADDITIONAL QUANTITATIVE EVALUATION

### C.1 SELF-SUPERVISION VS NOISY SUPERVISION

To evaluate the benefit of using the self-context, we design an experiment where we consider a supervised version as opposed to the self-supervised one, presented in this paper. Specifically, we use the vision encoder of CLIP ViT-B/32 (Radford et al., 2021) to label an image captioning dataset, namely the CC3M dataset (Sharma et al., 2018). Then, we use the ImageNet1k labels and encode them with the text encoder of CLIP ViT-B/32. We perform zero-shot classification of the CC3M images, and we consider this as a noisy supervised labeling of the dataset. Note that this is different from the self-context version that we propose, since we perform self-supervised labeling of the clustered images, without using any direct supervision. Surprisingly, our self-supervised variant outperformed the noisy supervised one across the majority of few-shot settings, which shows the effectiveness of the self-supervised signal.

Table 2: Comparison between the proposed self-context variant and a noisy supervised one, evaluated on the Real-Name miniImageNet dataset in accuracy(%). The SeCAt approach benefits from performing the clustering step as a way to learn a self-context.

| | 2-way | | | 5-way | | |
|---|---|---|---|---|---|---|
| variant | 1-shot | 3-shot | 5-shot | 1-shot | 3-shot | 5-shot |
| noisy supervision | 76.9 | 81.3 | 73.7 | 55.2 | **65.0** | **64.2** |
| self-supervision | **85.7** | **88.2** | **83.2** | **68.6** | 59.3 | 58.0 |

## C.2 SELF-SUPERVISED DATA GENERATION FOR PSEUDO ANNOTATIONS IN INFERENCE

In this experiment, we explore the potential of using self-supervised data generation to create pseudo annotations for inference purposes. To do this, we reassign labels in the Real-Name miniImageNet dataset based on our predetermined k-means clustering names and then evaluated the model's performance. Results for the 2-way 1-shot and 5-way 1-shot few-shot tasks are shown in Table 3. It reveals that

| Methods | 2-ways 1-shot | 5-ways 1-shot |
|---|---|---|
| ClipCap | 0.0 | 0.0 |
| Frozen | 33.7 | 14.5 |
| FROAMGe | 31.0 | 17.5 |
| **SeCAt (Ours) w/ pseudo** | 77.3 | **69.7** |
| **SeCAt (Ours)** | **85.7** | 68.6 |

Table 3: SeCAt enables self-supervised data generation to construct the pseudo annotations for inference, provides competitive performance compared with original SeCAt, and outperforms FROMAGe.

SeCAt effectively categorizes images within its context, even without using the original labels. Instead, SeCAt leverages the labels associated with the cluster centroids from the fine-tuning stage. Notably, its performance exceeds that of both OpenFlamingo and FROMAGe.

## C.3 UNSUPERVISED RETRIEVAL-BASED AUGMENTATION IN SECAT

In this experiment, we investigate the augmentation of the context samples presented to the model alongside a query image. We begin by sampling 2-ways 1-shot few-shot learning tasks from the Real-Name miniImageNet dataset. Then, we enhance their context samples by adding four additional examples per class from data clustered using k-means, thus transforming the tasks into 2-way 5-shot settings. The performance of SeCAt in this augmented setting is shown in Table 4. Notably, even with 80% of the context samples sourced from our clustered data, the results are close with the orig-

| Methods | 2-ways 5-shots |
|---|---|
| ClipCap | 0.0 |
| Frozen | 66.0 |
| FROMAGe | 50.4 |
| **Augmented-SeCAt (Ours)** | 74.0 |
| **SeCAt (Ours)** | **83.2** |

Table 4: SeCAt enables unsupervised retrieval-based augmentation, provides competitive performance compared with original SeCAt, and outperforms FROMAGe.

inal SeCAt approach and outperform FROMAGe. This emphasizes the potential of using constructed clusters to enrich context samples without the need for supervised data.

## C.4 PER-DATASET EVALUATION ON THE HARD-SPLIT

The evaluation performed on the hard-split is done by aggregating five different datasets, namely *OxfordPets* (Parkhi et al., 2012), *Flowers102* (Nilsback & Zisserman, 2008), *Food101* (Bossard et al., 2014), *CUBS-200* (Wah et al., 2011) and *SUN397* (Xiao et al., 2010). In this section, we analyze the performance of our SeCAt-trained model on the separate datasets and we report the results in Figure 2, across 2- and 5-ways, each with 1- and 5-shots. Similarly, as in the aggregated dataset version, we can observe better performance by SeCAt when using fewer shots. In particular, our approach shows better performance on 17 out of 20 few-shot settings, compared to the $5\times$ larger FROMAGe model (Koh et al., 2023). Note that FROMAGe shows the near-zero performance of the

*SUN397* dataset, across all few-shot settings, demonstrating the lack of appropriate pre-training data for scene understanding in this case. Unlike this, our SeCAt approach does not entirely rely on the pre-training data, since it is able to leverage the image-caption mappings from the context to make the prediction. However, when we increase both the ways and shots, OpenFlamingo (Awadalla et al., 2023) shows better performance, which is understandable given the scale of its language backbone (9B parameters).

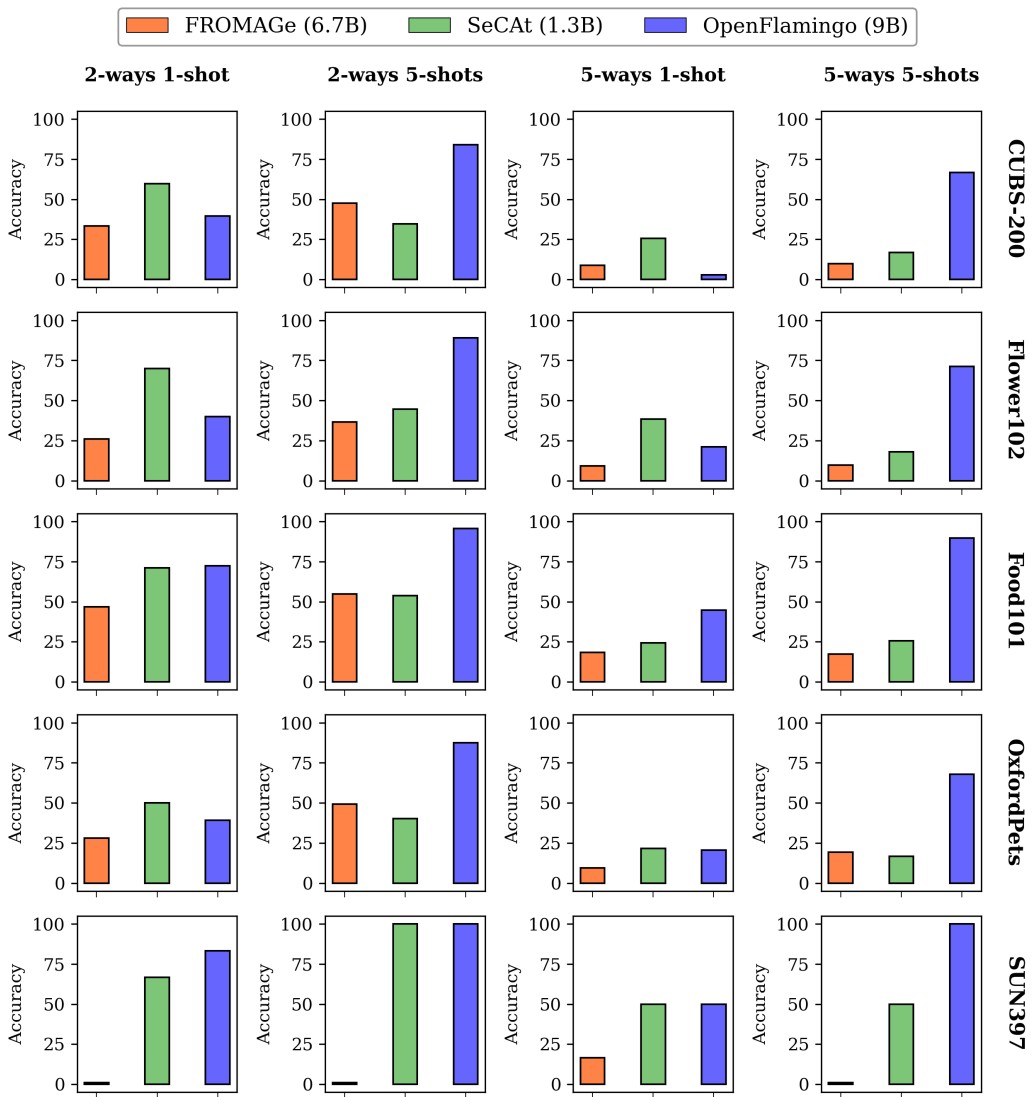

Figure 2: Performance evaluation per-dataset on different few-shot settings, namely the 2- and 5-way each with 1 and 5-shots from the Hard-split in accuracy(%). Note that OpenFlamingo (Awadalla et al., 2023) is considered an upper-bound. Results are consistent with the aggregated split. Our SeCAt-trained model shows better or comparable performance w.r.t the 5× larger FROMAGe model.

# D ADDITIONAL QUALITATIVE EVALUATION

In the next section, we provide additional qualitative comparisons between SeCAt-trained model and two other baselines, namely ClipCap (Mokady et al., 2021) and FROMAGe (Koh et al., 2023). We show a few successful cases in Figure 3 and also failure cases in Figure 4. We use GPT-Neo as a language model backbone across all examples. From Figure 3 it can be noticed that ClipCap produces

misleading and somewhat nonsense captions, showing it is not capable of digesting a multimodal context. This is due to the fact that it has not been trained to observe interleaved sequences of images and captions and uses a small language backbone that is incapable of in-context learning. On the other hand, FROMAGe shows better qualitative performance than ClipCap in terms of generating meaningful words. However it tends to perform standard image captioning, and it generates verbose and long captions, instead of following the context samples. This shows that FROMAGe relies on its pre-training knowledge for caption generation rather than learning directly from the context. We also show a few failure cases of our SeCAt-trained model in Figure 4. It can be seen that our predictions sometimes are missing particular tokens to have the complete correct words, such as *ing bowl* instead of *mixing bowl*.

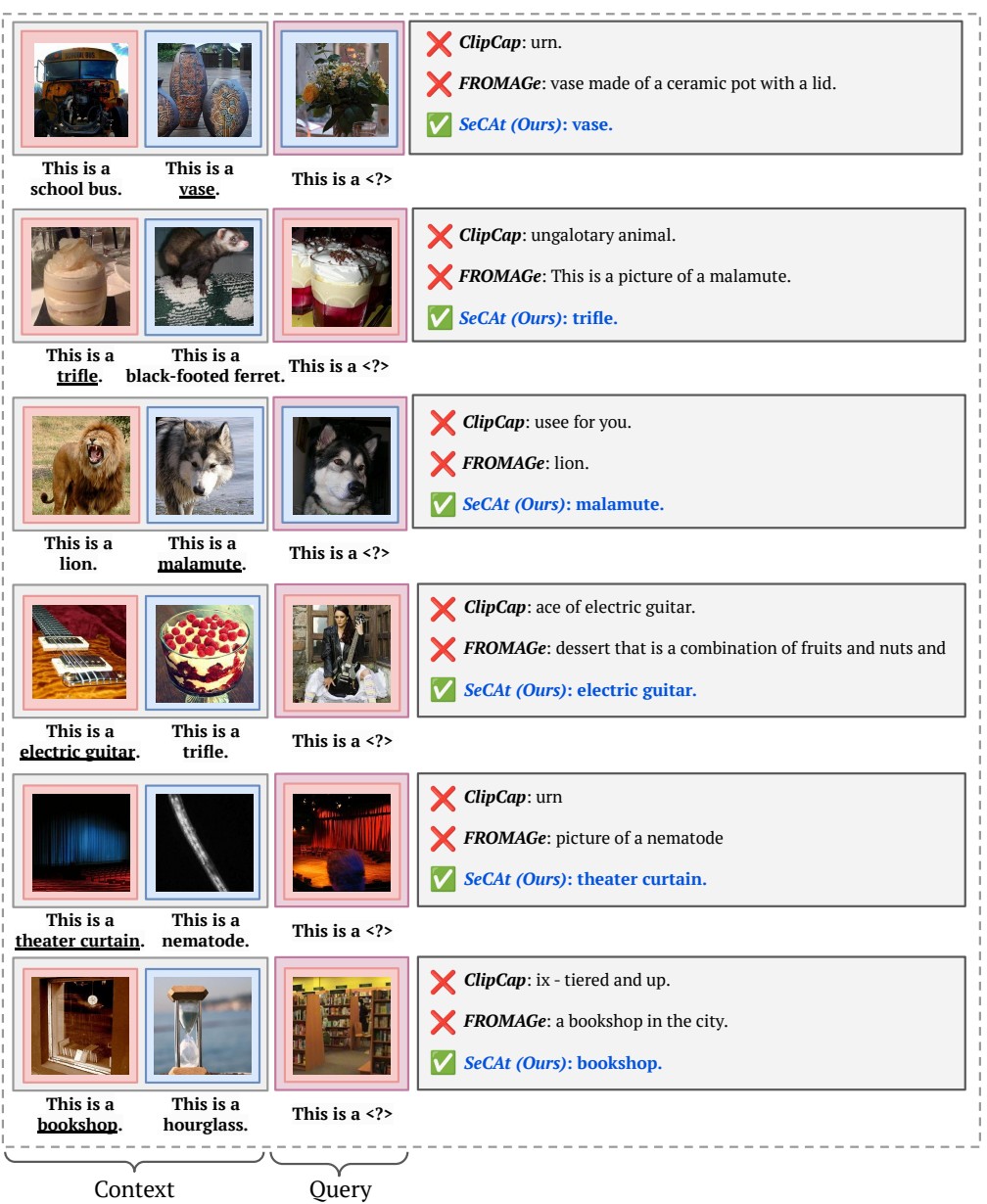

Figure 3: Qualitative comparison between SeCAt-trained model and two other baselines, namely ClipCap (Mokady et al., 2021) and FROMAGe (Koh et al., 2023), on a 2-way 1-shot task from Real-Names miniImageNet, showing successful cases of SeCAt.

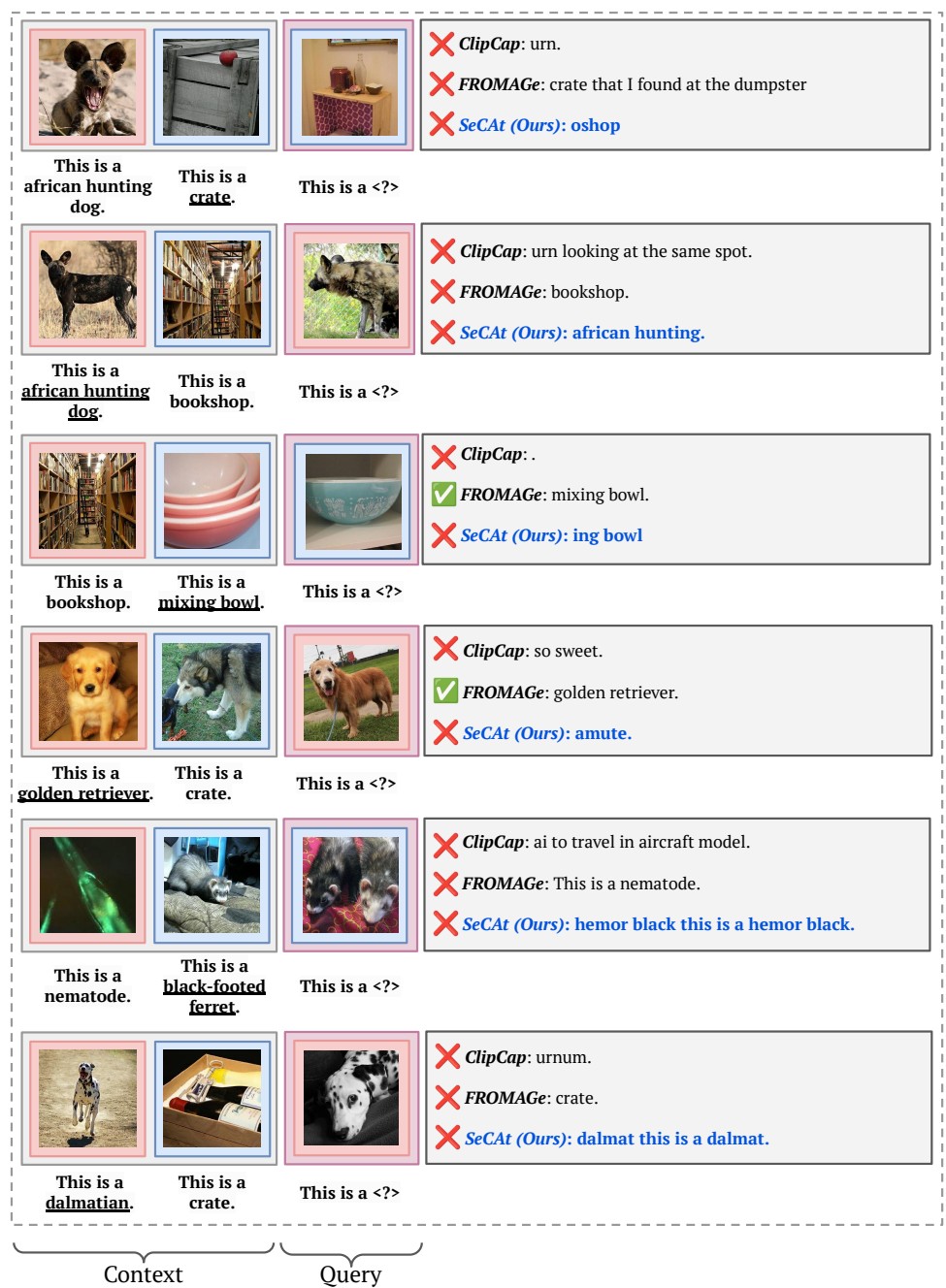

Figure 4: Qualitative comparison between SeCAt-trained model and two other baselines, namely ClipCap (Mokady et al., 2021) and FROMAGe (Koh et al., 2023), on a 2-way 1-shot task from Real-Names miniImageNet, showing failure cases of SeCAt.