# OpenReview forum: "Small Visual Language Models can also be Open-Ended Few-Shot Learners"
_ICLR.cc/2024/Conference — ICLR 2024 Conference Withdrawn Submission_

### Official Review · Reviewer_uhMd · 2023-10-28

**Soundness:** 2 fair
**Presentation:** 3 good
**Contribution:** 1 poor
**Rating:** 3
**Confidence:** 4

**Summary:**

This paper proposes a self-supervised training method for vision language models by assigning pseudo-class labels to images. The training process imitates the in-context learning setting by creating interleaved image-"caption" pairs where the caption is the assigned pseudo-class name. Through this training method, the 1B scale vision-language model outperforms other large-scale vision-language models on a series of fast concepts binding benchmarks.

**Strengths:**

1. It is an interesting and important topic to explore the potential few-shot learning ability on small-scale vision language models.
2. The overall method is simple and clear.

**Weaknesses:**

The main problem of this paper is that their design and experiments can not support the claim that the proposed model has the few-shot or in-context learning ability.

Few-shot learning for vision language models requires the model to first infer the underlying task according to the context and give the accordingly response. The paper only includes the fast concept binding task which exactly matches the proposed training paradigm as an evaluation for the few-shot learning ability. The proposed training method only teaches the model to pick one "category" name from the given context based on the visual similarity between the query image and context images instead of learning general multi-modal abilities. Therefore, it is foreseeable to have good performance on those fast concept binding tasks compared with other real vision-language models. The authors should test the model on different tasks (eg. VQA) in the few-shot setting to validate its claim. Otherwise, the position of this paper should be reconsidered and might need to be compared with meta-learning methods.

**Questions:**

When evaluating other vision-language models, is the task induction included (eg. Answer with ClassA or ClassB)?

---

### Official Review · Reviewer_1gge · 2023-10-31

**Soundness:** 3 good
**Presentation:** 2 fair
**Contribution:** 3 good
**Rating:** 8
**Confidence:** 4

**Summary:**

The authors present SeCAt, a self-supervised approach for few shot image captioning that leverage small visual language models.
By relying on psedo-labeling and subsequent matching they are able to leverage vast unannotated datasets defining a "self-context" of interleaved image and pseudo-captions with the goal to tune the model for generating the right pseudo-caption.
They show how this approach can outperform larger models in a variety of few shot tasks complementing the experiments with a series of ablation studies that analyse the impact of various techniques applied in SeCAt.

**Strengths:**

They authors define an effective recipe for few-shot visual language models alignment that boosts performance of small models.
The introduction of the "self-context" is an extremely interesing concept as it can be controlled arbitrarily to match granularity/complexity of the task at hand.
Kudos for the plan to release the code.

**Weaknesses:**

The concept of k-ways should be explained better when it is introduced. I found myself having to go back multiple times while reading the work.
The quantitative comparisons are missing the strongest baseline (OpenFlamingo, the upper bound). For completeness and transparency it should be included.

**Questions:**

How does the approach scale with the long contexts? Is it straightforward to extend it to arbitrarily long "self-contexts"?
Can you explain why SeCAt exhibit the weird trend of decreasing performance when incrementing the number of shots? This is something quite surprising based on the trend exhibited usually, and confirmed by all the baselines.

---

### Official Review · Reviewer_4XKc · 2023-10-31

**Soundness:** 2 fair
**Presentation:** 3 good
**Contribution:** 2 fair
**Rating:** 5
**Confidence:** 4

**Summary:**

This paper proposes a method called SeCAt to enable smaller-scale vision-language models to exhibit few-shot learning capabilities. In particular, they utilize pretrained vision and language models and aim to continue training (finetuning) the language model component using interleaved text-image pairs in context, following the methodology of previous work like Flamingo, but with modifications that are hypothesized to be important for unlocking few-shot capabilities in smaller models. Specifically, they propose a technique that pseudo-labels images (using arbitrary nouns that don’t necessarily semantically correspond to the content of the image), and creating captions accordingly, with a template like “an image of a X” where X is replaced with the assigned noun for that image. The way that this image pseudo-labeling is performed is by first clustering images in the embedding space of a vision encoder using k-means. Then, a random subset of vocabulary words is chosen, their language embeddings are computed, and the Hungarian matching algorithm is used to match one word per image cluster, via the similarity between that word embedding and each cluster prototype. Finally, once these matched image-caption pairs are formed, they are used to create interleaved text-image “self-contexts” for few-shot learning tasks for finetuning the LM. These have the usual format of a support set (each support image followed by its caption), followed by a query image, and the language model is finetuned to predict the correct caption for that query image, when prompted with the self-context. At inference time, the model is kept entirely frozen and few-shot learning tasks are solved via in-context learning. The authors consider simple few-shot learning tasks in datasets created by mini-imagenet, following some previous work. They find empirically that, for these few-shot tasks, the proposed approach applied on a smaller language model is able to surpass the performance of larger language models. They also conduct a thorough set of ablations to analyze the influence of different components and design choices of their method.

**Strengths:**

- This paper investigates a question that is both scientifically interesting and practically important: is large scale necessary for emerging few-shot learning capabilities in VLMs?
- The paper is well-written and easy to follow for the most part
- It is an interesting finding that smaller language models can match or exceed the performance of larger ones on some tasks

**Weaknesses:**

- Some related work is missing. The authors should cite work that attempts to automatically construct few-shot learning tasks without supervision / class labels from the ‘older’/’traditional’ few-shot literature too (e.g. in the context of few-shot classification on vision tasks) and newer. Some examples are [A, B] (see references below, and several additional references therein).

- The few-shot learning problems used for evaluation are quite simple. It would be great to apply this approach on other multi-modal few-shot learning tasks from related work (e.g. Flamingo)

- I’m a bit concerned that the few-shot binding task in particular is one that the proposed method is particularly suited for, and thus the reported results may be more optimistic than they may have been for different few-shot learning tasks. The reason I think this is that the particular way in which self-contexts are formed for finetuning can be thought of as ‘binding’ visual concepts to new (random / unrelated) nouns. So the proposed approach can be viewed as finetuning the LM specifically on ‘binding few-shot tasks’ and thus as being ‘tailor-made’ for such tasks.

- The motivation of the proposed pseudo-labeling and self-context creation of SeCAt should be strengthened. Why is it better than finetuning on actual paired image-text as is done in previous works? If I understand the argument correctly, the point is to simulate *any* new concept, and teach the model to in-context learn by figuring out to match ‘symbols’ to new items, rather than recalling known semantic relationships between words and visual concepts. And the argument is that this is more important for smaller models? It would however greatly strengthen the motivation of the current approach to actually demonstrate empirically the failure of using ‘standard’ interleaved text-image finetuning versus their proposed method (in the context of exactly the same small language models). It would be very interesting to investigate how the relative performance of those two (usual interleaved text-image finetuning versus SeCAt) changes as the size of the model increases. Please let me know if this is already one of the baselines and I’m missing it.

- Related to the above point, for the ablation in Table 3b, aside from ‘nonsense’, ‘numbers’ and ‘nouns’, an additional row could be for assigning the actual ‘matching’ noun. The implicit hypothesis if I understand correctly is that that would be worse, but it would be good to examine this empirically.

- For ablation 3e that studies the impact of the model size on few-shot learning capabilities, there is an important aspect that isn’t studied, namely, how much does adding SeCAt *improve* the few-shot learning capabilities of models of different sizes (that is, plotting the relative improvement of few-shot learning in the ‘raw’ model versus after SeCAt finetuning). And ideally, tying this in with what I wrote above, this would also be compared to the relative improvement that would have been obtained by more ‘standard’ interleaved text-image finetuning.


References
=========
- [A] Unsupervised Learning Via Meta-Learning. Hsu et al. ICLR 2019.
- [B] Unsupervised Meta-learning via Few-shot Pseudo-supervised Contrastive Learning. Jang et al. ICLR 2023.

**Questions:**

- for the qualitative analyses in Figure 3, why is it worse to be more verbose as FROMAGe is? If I understand correctly, the model wasn’t prompted in a way that asks it to be concise, so it’s not surprising that, unless finetuned exactly for this task format, the generated outputs may be more verbose / have a different structure (but it seems to me that that doesn’t make them incorrect).

- since table 3f does show some amount of overfitting to the prompt template, why not also vary this during training (just like the difficulty is varied and the ‘shot’ is also varied)? Did the authors consider this?

- when discussing the finding that larger models are able to in-context few-shot learn well, the authors say (in the intro) that this is attributed to their increased parameter count as well as the fact that they have been trained with interleaved text-image pairs. Has there been work trying to disentangle those two? E.g. reporting results on increasingly large models without the interleaved finetuning? If so, it would be useful to summarize those findings in the intro too.

- Why is it that the authors finetune the language model but not the vision model during SeCAt training? Were other options explored (finetuning only vision or both?)

---

### Official Review · Reviewer_t52H · 2023-11-01

**Soundness:** 2 fair
**Presentation:** 2 fair
**Contribution:** 2 fair
**Rating:** 3
**Confidence:** 4

**Summary:**

This paper presents a simple method called self-context adaptation (SeCAt), which make a small visual language models to be a powerful few-shot learner. Specifically, based on the clustering of images and randomly assigned category name, SeCAt simulate the multi-modal in-context learning.

**Strengths:**

(1) The paper is easy to follow and well-written. Especially, figures are well-designed and ideas are well presented.

(2) Empirical results show the strong performance of the proposed method. They provide sufficiently good ablation studies to deeper understand the logic behind the performance gain.

**Weaknesses:**

While the paper is interesting to read, I have several major concerns about novelty and effectiveness of the method.

(1) Using the unrelated name to improve the in-context learning ability is not new in language model [1]. While the domain is slightly different, author need to discuss the difference with the study and clarify the contribution.

Especially, according to [1], they show that symbol tuning does not improve performance on small scale model (with relevant labels) does not improve performance, which is somewhat contradicting with the claim in this paper.

[1] Symbol tuning improves in-context learning in language models


(2) While the paper claim that it is open-ended, the evaluated task is limited to the standard few-shot classification task. In other words, the trained model is strongly biased toward generate shorter sentences compared to the existing methods like Frozen and FROMAGe, which are not designed to just solve classification.

In that sense, I think author should compare the method with the standard few-shot baseline, rather than just comparing with the pre-trained vision language models.

**Questions:**

see weakness part